

# Advancing brain tumor detection: harnessing the Swin Transformer's power for accurate classification and performance analysis

Abdullah A. Asiri[1], Ahmad Shaf[2], Tariq Ali[2], Muhammad Ahmad Pasha[2], Aiza Khan[2], Muhammad Irfan[3], Saeed Alqahtani[1], Ahmad Alghamdi[4], Ali H. Alghamdi[5], Abdullah Fahad A. Alshamrani[6], Magbool Alelyani[7] and Sultan Alamri[4]

[1] Radiological Sciences Department, College of Applied Medical Sciences, Najran University, Najran, Saudi Arabia
[2] Department of Computer Science, COMSATS University Islamabad, Sahiwal Campus, Sahiwal, Pakistan
[3] Faculty of Electrical Engineering, Najran University, Najran, Saudi Arabia
[4] Radiological Sciences Department, College of Applied Medical Sciences, Taif University, Taif, Saudi Arabia
[5] Department of Radiological Sciences, Faculty of Applied Medical Sciences, University of Tabuk, Tabuk, Saudi Arabia
[6] Department of Diagnostic Radiology Technology, College of Applied Medical Sciences, Taibah University, Taibah, Saudi Arabia
[7] Department of Radiological Sciences, College of Applied Medical Science, King Khalid University, Abha, Saudi Arabia

Corresponding author
Ahmad Shaf,
ahmadshaf@cuisahiwal.edu.pk

## ABSTRACT

The accurate detection of brain tumors through medical imaging is paramount for precise diagnoses and effective treatment strategies. In this study, we introduce an innovative and robust methodology that capitalizes on the transformative potential of the Swin Transformer architecture for meticulous brain tumor image classification. Our approach handles the classification of brain tumors across four distinct categories: glioma, meningioma, non-tumor, and pituitary, leveraging a dataset comprising 2,870 images. Employing the Swin Transformer architecture, our method intricately integrates a multifaceted pipeline encompassing sophisticated preprocessing, intricate feature extraction mechanisms, and a highly nuanced classification framework. Utilizing 21 matrices for performance evaluation across all four classes, these matrices provide a detailed insight into the model's behavior throughout the learning process, furthermore showcasing a graphical representation of confusion matrix, training and validation loss and accuracy. The standout performance parameter, accuracy, stands at an impressive 97%. This achievement outperforms established models like CNN, DCNN, ViT, and their variants in brain tumor classification. Our methodology's robustness and exceptional accuracy showcase its potential as a pioneering model in this domain, promising substantial advancements in accurate tumor identification and classification, thereby contributing significantly to the landscape of medical image analysis.

# INTRODUCTION

Cancer is one of the most studied diseases. Among all the other types of cancer, brain tumors are considered a highly studied type of cancer (*Tiwari et al., 2022*). The rate of brain tumors in humans is about 250,000 per year with about 2% of malignancies. Brain tumors appear with different kinds of symptoms related to age, abnormality, and mental circumstances. The infinite cell divisions inside the brain produced the tumor (*Bhanothu, Kamalakannan & Rajamanickam, 2020*). Primary tumors and metastatic tumors are two major types of brain tumors. Primary brain tumors could be spotted anywhere in the brain, but do not have locomotive ability. While metastatic brain tumor initiates itself as a cancer that could be found anywhere in the body and then it leads the way to the brain. We can classify primary brain tumors into subgroups *i.e.,* malignant brain tumors and benign brain tumors (*Çınarer & Emiroğlu, 2019*). We can track down malignant tumors by the techniques of image processing and algorithms responsible for the classification. Diagnostics of brain cancer can be done in the form of invasive or non-invasive. Invasive approaches include tumor sampling and biopsy techniques, These techniques have been used as a gold standard for the diagnosis of cancerous cells by observing features of cells for malignancy conformation under a microscope. The non-invasive method required the thorough evaluation and scanning of the brain utilizing various techniques of imaging. CT (computed tomography), MRI (magnetic resonance imaging), and X-ray techniques are safer and faster ways to diagnose cancer instead of biopsy. Image modalities help radiologists to figure out brain disorders in surgical progressions.

CAT (computer-assisted tools) MRI (magnetic resonance imaging) has led computing machines and helps to decrease hardware costs for cancer diagnosis. Different CAT-based methods have been introduced *i.e.,* machine learning (ML) and deep learning (DL) for automatic tumor segmentation (*Tandel et al., 2019*). To classify images obtained from an MRI of the brain, uncontrolled techniques are used in combination with classifiers *i.e.,* artificial neural networks (ANN), and support vector machines (SVM). In this context, a supervised classification method has been proposed *i.e.,* the K-NN method. Unsupervised neural networks (UNN) have been introduced for the classification of results obtained from brain MRI images. In, the hybrid technique discussed the results in two different ways *i.e.,* normal and abnormal, with the machine learning (ML) algorithm named SVM. Two methods *i.e.,* BPNN and KNN classifiers. While in terms of accuracy, 70% accuracy has been shown by KNN and 72.5% by BPNN. An accuracy of 80% has been observed by using tissue analysis to classify low and high grades of glioma heterogeneity (*Singh & Kaur, 2012*). Based on the imaging features a multivariate estimation model has been proposed with an accuracy of 74% (*Jafari & Shafaghi, 2012*). In, 90% accuracy has been recorded for the classification of non-tumor MRI using the random forest classifier (*Sudharani, Sarma & Prasad, 2016*).

DL is often preferred over traditional machine learning approaches for the detection of brain tumors as it is capable of automatically learning and obtaining meaningful features from input. Various other features include feature learning, hierarchical representations, handling high-dimensional data, and generalization. DL has gained significant traction in brain image analysis across various applications, including the classification of normal or abnormal brain tumors, segmentation of different regions (such as edema), and Alzheimer's diagnosis (*Litjens et al., 2017*). Among DL models, the convolutional neural network (CNN) stands out as the most utilized for medical image classification.

CNNs excel in capturing the spatial relationships between pixels in a way of hierarchy. This is accomplished through the application of learned filters that convolve the images, thereby constructing hierarchical manners of feature maps (*Zhou, He & Jia, 2020*). By employing multiple layers of convolution, the resulting features exhibit translation and distortion invariance, leading to a high level of accuracy (*Biswas et al., 2018*). The image processing is efficiently done by using CNN. Four transfer learning models are used including ResNet-50, Inception V3 VGG16, and Mobile Net to analyze the brain tumor by using MRI dataset (*Kumar, Dabas & Godara, 2017*). An automatic classification model has been proposed (*Ramteke & Monali, 2012*). The results were generated by a CNN classifier with 80% accuracy. An accuracy of 82.49% has been produced by *Graves, Mohamed & Hinton (2013)* on medical images. In a recent study, the authors incorporated transfer learning from pre-trained models such as VGG16, VGG19, ResNet50, and DenseNet21 using various optimization algorithms. After thorough analysis, the authors found that ResNet50 demonstrated the highest performance compared to the other models (*Polat & Güngen, 2021*). Nevertheless, a drawback of CNNs is their inability to effectively grasp long-range data or dependencies, chiefly ascribed to their diminutive kernel size (*Hatamizadeh et al., 2021*). Long-range dependencies refer to situations where the desired output is influenced by image sequences that are presented at distant time points. In medical images, visual representations often exhibit a sequential organization due to the human organs' similarities (*Tan et al., 2023*). The destruction of these sequential dependencies can have a substantial effect on the efficiency of CNN models. This is because the dependencies existing between the properties of image sequence including patch, modality, and slice hold valuable information that contributes to the overall understanding and analysis of the images (*Dai, Gao & Liu, 2021*).

Techniques capable of processing sequence relations are effective in handling these long-range dependencies. A specialized type of transformer model, called Vision Transformer (ViT), is designed for image analysis purposes. Notably, the ViT showcased superior performance compared to CNN models, particularly when trained on the JFT dataset containing a massive collection of 300 million images (*Dosovitskiy et al., 2021*). ViT architecture is built upon the foundation of the vanilla Transformer which has garnered significant attention recently due to excellent performance in machine translation and NLP tasks. The mechanism of self-attention utilized in ViT plays a crucial role in modeling such dependencies, which is particularly valuable for precise segmentation of the brain tumor. Global and local featured learning can be done by combining ViT-based models and token embeddings, allowing them to effectively capture and leverage the information

**Table 1  Difference between ViT and Swin Transformer.**

| Feature | ViT | Swin Transformer |
|---|---|---|
| Self-attention mechanism | Global | Window-based |
| Computational complexity | High | Low |
| Efficiency | Low | High |
| Performance | Good | Better |

contained in long-range dependencies (*Raghu et al., 2021*). ViT has shown promising performance across various benchmark datasets (*Wang et al., 2021*). Attention-based transformer networks have gained significant prominence in natural language processing tasks in recent times (*Touvron, Cord & Jégou, 2022*).

In the ViTs, the encoder module is specifically utilized for performing the classification of images. It achieves this by mapping a sequence of image patches to their corresponding semantic labels. Unlike traditional CNN architectures that primarily employ filters with a limited local receptive field, the ViT leverages the power of the attention mechanism (*Girdhar et al., 2023*). This allows the model to analyze images based on regions and collect data images, enabling a more comprehensive understanding and analysis of the visual content. Despite high efficiency and significant results, ViTs also have some limitations (*Kenton & Toutanova, 2019*). The main limitation of the research is its necessity for a large amount of labeled data (*Dai et al., 2021*). Due to various parameters and for satisfactory performance, labeled training data is needed. But this task is too expensive and time-consuming. One of the ways to overcome this limitation is to adapt the pre-trained ViT models. This includes large-scale ViT like ImageNet (*Cha et al., 2022*). It trains the ViT on a large dataset in a semi-supervised manner. By using smaller datasets, the pre-trained ViT models help the model to give a competitive performance with a limited labeled dataset (*Weng et al., 2022*). Using a pre-trained model and fine-tuning it according to the need, will save computational resources and time. This is an effective approach for various tasks including segmentation, object recognition, and image classification.

Several studies have discovered the use of ViT models for dense vision tasks like the detection of objects and segmentation. However, these approaches have exhibited comparatively lower performance, often employing direct up-sampling or deconvolution techniques (*Tournier et al., 2008*). Meanwhile, other works have attempted to enhance image classification by modifying the ViT architecture (*Jiang et al., 2021*). ViT uses a global self-attention mechanism, which means that it computes the attention between all pairs of tokens in the input sequence. This can be computationally expensive, especially for large images. The shifted window partitioning approach of the Swin Transformer addresses this issue by computing self-attention only within each local window. This reduces the computational complexity and makes the Swin Transformer more efficient. Table 1 summarizes the key differences between the ViT and Swin Transformer architectures.

In the proposed work, the Swin Transformer architecture introduces a novel approach to image analysis by utilizing a patch-splitting module to segment input RGB images into non-overlapping patches, each treated as a "token". These tokens are then embedded into a user-defined feature size. The architecture incorporates stages with patch merging layers that progressively reduce token count, forming a hierarchical representation while maintaining resolution. Crucially, a shifted window partitioning approach alternates between regular and shifted window strategies, enhancing cross-window connections without sacrificing efficiency. While inspired by the Vision Transformer, it also addresses its limitations, such as quadratic complexity in global self-attention, by employing a window-based self-attention strategy. The architecture's Swin Transformer block integrates a Multi-Head Self-Attention (MSA) module with GELU-activated MLPs, introducing inter-window connectivity and information flow. This window-based approach significantly enhances computational scalability. Collectively, these components endow the Swin Transformer with the capacity to capture intricate patterns in image data, positioning it as a promising alternative to traditional convolutional neural networks and mitigating certain limitations of the Vision Transformer.

This article is organized as follows: detailed dataset description and the proposed Swin model architecture is given in 'Material and methodology', experimental results and discussion are given in 'Result and Discussion', and finally, a conclusion is given in 'Conclusion'.

## MATERIAL AND METHODOLOGY

We sourced this dataset from Kaggle, a well-known online platform recognized for its extensive collection of high-quality datasets and resources for machine learning and data science. Anyone can access and download the specific dataset we used for this research from the following link: https://www.kaggle.com/datasets/sartajbhuvaji/brain-tumor-classification-mri/versions/2. The dataset's comprehensive nature, along with its well-defined class labels, provides a solid foundation for training and evaluating our proposed Swin Transformer model for the crucial task of multi-class brain tumor detection and classification on MRI images as shown in Fig. 1. The dataset utilized for this research consists of 2,870 MRI images depicting the human brain. These images were carefully selected and divided into four distinct classes, representing different brain-related conditions. In the dataset, the pituitary class (827 images), which encompasses glandular tumors; the glioma class (826 images), representing glial cell tumors; the no-tumor class (395 images), indicating the absence of tumors; and the meningioma class (822 images), includes tumors originating from the protective membranes surrounding the brain. To ensure a robust evaluation of our proposed model, we split a dataset containing 2,870 images across four classes into training and testing sets. It divides the dataset into train images and test images using an 85/15 ratio. Additionally, it further divides the dataset into a train set and a validation set with an 80/20 ratio for training and validation purposes, respectively. Both splits use a random state of 42 for reproducibility. For training the model, we allocated 2,296 images, allowing it to learn the intricate patterns and characteristics

| glioma_tumor | no_tumor | meningioma_tumor | pituitary_tumor |
|---|---|---|---|
| 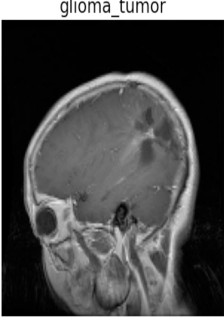 | 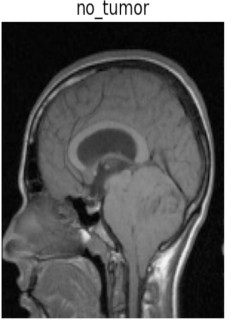 | 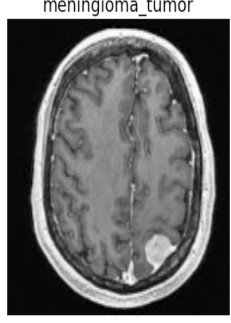 | 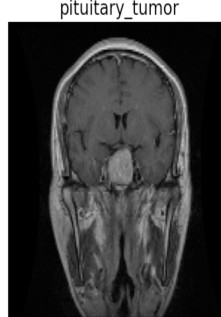 |

**Figure 1** Sample image from each label of the dataset.

associated with various brain tumor types. For an unbiased assessment of the model's performance, we reserved 574 images as a validation set and 431 as a testing set. This test set served as a benchmark to evaluate how well the model generalizes and performs on previously unseen data.

## Preprocessing steps

Data preprocessing holds immense importance in machine learning pipelines. It lays the foundation for model success by shaping raw data into a format that is conducive to effective learning. The following preprocessing steps were involved.

### Data gathering

Four directories are defined, each containing images for a specific type of brain tumor or healthy brain tissue. A loop iterates through these directories, listing all the image files within each. File paths and corresponding labels (tumor type or "no tumor") are appended to separate lists.

### Data preprocessing

Pandas Series objects are created for file paths and labels. These series are combined into a single DataFrame (*tumor_df*). The first five rows of the data frame are printed to get a glimpse of the data. Class distribution across labels is printed using *value_counts()* to understand the balance of classes. The data frame's shape is obtained to reveal the total number of data points and features.

### Train-test-validation split

Scikit-learn's *train_test_split* function is used to split the data into separate training, testing, and validation sets. The training set is further split into training and validation subsets using the same function.

### Image data augmentation

Keras Image Data Generator is used for data augmentation and image preprocessing. The mobilenet version 2 preprocess function is specified for preprocessing, which involves mean subtraction and scaling specific to MobileNet V2 architecture. Three separate image data generator instances are created for training, testing, and validation sets.

Each instance is configured with:

- Data source (data frame, *x_col* for file path, *y_col* for label)
- Target image size (both width and height resized to 128 pixels)
- Color mode ("RGB" for color images)
- Class mode ("categorical" for multi-class classification)
- Batch size (specifies the number of images processed together)
- Shuffle disabled (important to maintain order for cross-validation)

These steps prepare the data for training a proposed model for brain tumor classification. These preprocessing steps and their utilization in the proposed work has been presented in Algorithm 1.

## Swin Transformer architecture

In the Swin Transformer (Swin-T) architecture, the patch-splitting module is employed to partition the RGB input image into non-overlapping patches. Each patch is treated as a "token", and its properties are formed by merging or combining the pixels of the input RGB. This approach utilizes $4 \times 4$ dimensions, resulting in a feature dimension of 48 ($4 \times 4 \times 3$). Subsequently, a linear embedding process is applied to the features, mapping them to a user-defined size denoted as C. Multiple blocks within Swin-T incorporate a customized self-attention calculation, which is then applied to these tokens. The amalgamation of these blocks and the embedded system constitutes what is termed "stage 1". This stage ensures the uniformity of token count, specifically H * 4 $\times$ W * 4.

As the network's depth increases, patch merging layers are employed to progressively reduce the number of tokens, resulting in a hierarchical arrangement. Initially, the patch merging layer combines the attributes of neighboring patches in a $2 \times 2$ group. This is followed by a linear layer operating on the concatenated 4C-dimensional features, effectively downsampling the token count by a factor of $2 \times 2$, equivalent to a 2$\times$ down-sampling of resolution. The resulting output size is 2C. After the downsampling in "Stage 1", Swin Transformer blocks are employed to transform the features while preserving the resolution at H * 8 $\times$ W * 8. This stage, encompassing the initial patch merging and feature transformation, is denoted as "Stage 2". These steps are then reiterated twice more, known as "Stage 3" and "Stage 4", resulting in output resolutions of H * 16 $\times$ W * 16 and H * 32 $\times$ W * 32, respectively. By integrating these stages, a hierarchical representation is constructed with feature map resolutions comparable to conventional methods. As a result, the proposed architecture possesses the potential to function as a versatile alternative to the spine systems in existing approaches, yielding enhanced performance across a spectrum of vision objectives.

## Block of Swin Transformer

The Swin-T architecture presents an alteration to the conventional Transformer block by incorporating a Multi-Head Self-Attention (MSA) module founded on a shifted window approach. While the other layers within the Swin-T block remain unaltered, Fig. 2 illustrates. Specifically, the Swin Transformer block is structured with a shifted window-based MSA

---

**Algorithm 1:** Swin Transformer Image Classification

---

**Data:** Input data

**Result:** Output data

**Import** necessary libraries and packages

**Function** `Data_Preparation`

    **Define** class names class_names ← ['glioma', 'meningioma', 'no tumor', 'pituitary']

    class_names_label ← **CreateDictionary**(class_names)

    **Define** dataset directories tumor_directory ← "path/to/tumor_directory"

    **for** *directory* **in** *[class names]* **do**

        file_list ← **GetFilesInDirectory**(directory) **for** *file* **in** *file_list* **do**

            **Append** file **to** filepaths **if** *directory* **is** *glioma* **then**

                **Append** 'glioma_tumor' **to** labels

            **else if** *directory* **is** *meningioma* **then**

                **Append** 'meningioma_tumor' **to** labels

            **else if** *directory* **is** *pituitary* **then**

                **Append** 'pituitary_tumor' **to** labels

            **else**

                **Append** 'no_tumor' **to** labels

    **Created DataFrame**

**Function** *Swin_Transformer_Model_Setup*

    Hyperparameters and settings patch_size ← (2, 2)

    dropout_rate ← 0.03

    num_heads ← 8

    embed_dim ← 64

    window_size ← 2

    shift_size ← 1

    num_mlp ← 256

    num_patch_x ← **input_shape**[0] / patch_size[0]

    num_patch_y ← **input_shape**[1] / patch_size[1]

    learning_rate ← 1e-3

    batch_size ← 128

    num_epochs ← 10

    weight_decay ← 0.0001

    label_smoothing ← 0.1

    **Function** window_partition

    **Function** window_reverse

    **Function** DropPath

    **Function** WindowAttention

    **Function** PatchExtract

    **Function** PatchEmbedding

    **Function** PatchMerging

**Main**

`Data_Preparation Swin_Transformer_Model_Setup`

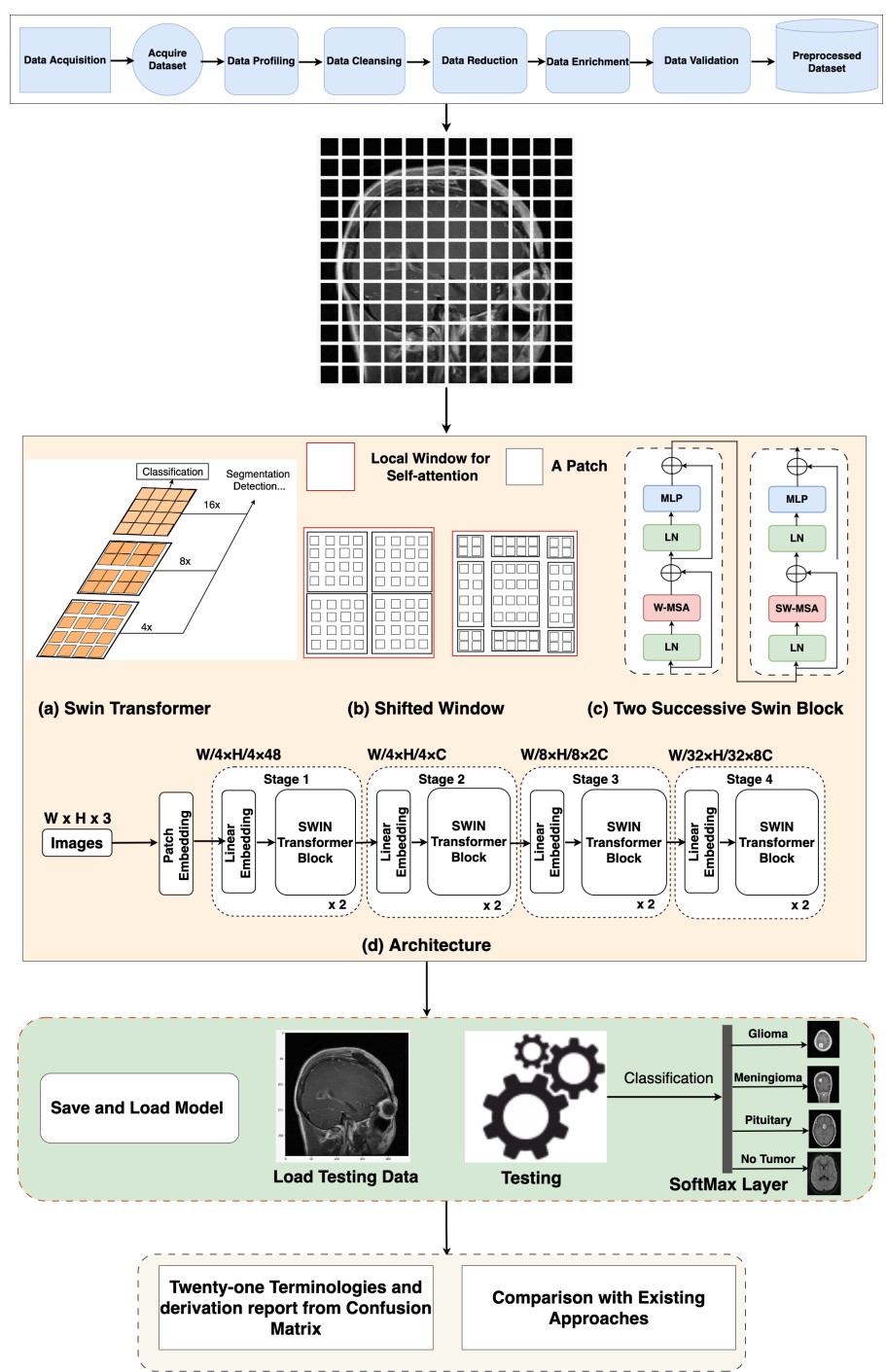

**Figure 2  Overall architecture of the proposed Swin Transformer.**

module, succeeded by a 2-layer multi-layer perceptron (MLP) with GELU non-linearity situated in-between. Layer normalization (LN) layers are applied to precede each MSA module and each MLP layer, and residual connections are introduced after each module to guarantee the preservation of information flow.

## Non-overlapping windows for self-attention

To address the computational inefficiencies of global self-attention, self-attention within localized windows has been used. These windows are organized in a non-overlapping fashion to achieve an even partition of the image. Each window is composed of $M \times M$ patches, with a typical default value of $M$ being 7. By adopting this window-based self-attention, the computational complexity transforms into a linear relationship with the number of patches, $h \times w$, in contrast to the quadratic complexity associated with global self-attention. This transition to a window-based approach enhances scalability and affordability, particularly in scenarios involving a substantial number of patches $(h \times w)$, where the computational demands of global self-attention become infeasible.

## Shifted window partitioning approach

To address the constraint of the window-based self-attention module, which does not establish connections across windows, an innovative solution known as the shifted window partitioning technique is used. This novel approach brings in cross-window connections while upholding the efficiency of non-overlapping windows. In successive Swin Transformer blocks, we adopt two distinct partitioning strategies alternately. The first module utilizes a regular window partitioning strategy, dividing the initial $8 \times 8$ feature map into $2 \times 2$ windows of size $4 \times 4$ $(M = 4)$. In the subsequent module, a shifted windowing configuration is adopted, where the windows are shifted from the previous layer's configuration. This alternating approach enhances the connectivity between windows and enables more comprehensive information exchange throughout the network. The displacement is achieved by moving the windows by $(bM/2, bM/2)$ pixels from the regularly partitioned windows. This shifted window partitioning approach allows for the computation of consecutive Swin Transformer blocks while incorporating cross-window connections. It enhances the modeling power of the architecture while maintaining efficient computations. The internal connection of the mathematical working of Swin-T:

$$z^l = \mathrm{MLP}\left(\mathrm{LN}(\hat{z}^l)\right) + \hat{z}^l \tag{1}$$

$$z^{l+1} = \mathrm{MLP}\left(\mathrm{LN}(\hat{z}^{l+1})\right) + \hat{z}^{l+1} \tag{2}$$

$$\hat{z}^l = \text{W-MSA}\left(\mathrm{LN}(z^{l-1})\right) + z^{l-1} \tag{3}$$

$$\hat{z}^{l+1} = \text{SW-MSA}\left(\mathrm{LN}(z^l)\right) + z^l \tag{4}$$

The standard Window-based, multi-head, and self-attention module (W-MSA) is a major part of the Swin Transformer block. The LN is present in the front, whereas S(W)-MSA is situated at the back. There are two GELU non-linearities in the last part of the block, that is, MPL. Due to this reason, this transformer has a multiple of two. In the above equations, the output of $f$ (S) W-MSA is indicated as $\hat{z}^l$, that of MPL is $z^l$ whereas $l$ is the position of the block of the Swin Transformer.

## Self-attention, multi-head and Window-based module

W-MSA first applies linear transformations to the input features, separating them into query, key, and value components. The attention scores are calculated by multiplying the query and key tensors and scaled by the square root of the head dimension. To introduce cross-window connections, a relative position bias is computed and added to the attention scores. This bias captures the relationships between different positions within the local windows. Additionally, if a mask is provided, it is incorporated into the attention scores to handle masked tokens. The attention scores are then softmax along the last dimension, followed by applying dropout. Finally, the value tensor is multiplied by the attention scores, and the resulting tensor is reshaped, projected, and subjected to dropout before being returned as the output of the self-attention and window-based module. Overall, the multi-head functionality is implicitly included in the Window Attention implementation. The attention scores are calculated independently for each attention head, allowing the model to capture different aspects of the input information. This helps enhance the model's representational capacity and allows for capturing diverse patterns and dependencies in the data.

To evaluate the model's performance, several parameters were used, which play a crucial role in classification algorithms: true positive (TP), true negative (TN), false negative (FN), and false positive (FP). These metrics are essential in assessing how well the model is performing in terms of correctly identifying tumor classes.

- **True positive rate (TPR):** The percentage of real positive cases that a classification model correctly classifies as positive.

$$TPR = \frac{TP}{TP + FN} \tag{5}$$

- **True negative rate (TNR):** It signifies the proportion of negative instances correctly classified as negative by a classification model.

$$(TNR) = \frac{TN}{TN + FP} \tag{6}$$

- **Positive predictive value (PPV):** It denotes the proportion of successfully predicted positive cases to all instances that a model accurately believed to be positive.

$$PPV = \frac{TP}{TP + FP} \tag{7}$$

- **Negative predictive value (NPV):** It signifies the percentage of correctly predicted negative cases among all instances that a model predicts to be negative.

$$\text{NPV} = \frac{\text{TN}}{\text{TN} + \text{FN}} \qquad (8)$$

- **False negative rate (FNR):** It represents the ratio of actual positive instances incorrectly classified as negative by a model.

$$\text{FNR} = \frac{\text{FN}}{\text{FN} + \text{TP}} \qquad (9)$$

- **False positive rate (FPR):** It indicates the ratio of genuine negative instances incorrectly predicted as positive by a model.

$$\text{FPR} = \frac{\text{FP}}{\text{FP} + \text{TN}} \qquad (10)$$

- **False discovery rate (FDR):** It represents the percentage of expected positive cases that are not positive.

$$\text{FDR} = \frac{\text{FP}}{\text{FP} + \text{TP}} \qquad (11)$$

- **False omission rate (FOR):** It signifies the percentage of real negative events that are mistakenly forecast as positive.

$$\text{FOR} = \frac{\text{FN}}{\text{FN} + \text{TN}} \qquad (12)$$

- **Positive likelihood ratio (LdRo+):** It is the link between the likelihood of a positive test result under specific conditions and the likelihood of a positive test result under different conditions.

$$\text{LdRo+} = \frac{\text{TPR}}{\text{FPR}} \qquad (13)$$

- **Negative likelihood ratio (LdRo-):** It is the ratio of the likelihood that a test result will be negative given the absence of a condition to the likelihood that the condition will be present.

$$\text{LdRo-} = \frac{\text{FNR}}{\text{TNR}} \qquad (14)$$

- **Prevalence threshold (PT):** It is the predictive probability level at which the positive predictive value matches the negative predictive value.

$$\text{PT} = \frac{\sqrt{\text{FPR}}}{\sqrt{\text{TPR}} + \sqrt{\text{FPR}}} \qquad (15)$$

- **Threat score (TS):** It is the product of the True positive rate and the positive predictive value, encapsulating the combined accuracy of positive predictions.

$$\text{TS} = \frac{\text{TP}}{\text{TP} + \text{FN} + \text{FP}} \qquad (16)$$

- **Prevalence (Pe):** It pertains to the proportion of the population (P) afflicted by the condition (N) under study.

$$Pe = \frac{P}{P+N} \tag{17}$$

- **Accuracy (AC):** It represents the proportion of correct predictions made by a model across all instances.

$$AC = \frac{TP+TN}{TP+TN+FP+FN} \tag{18}$$

- **Balanced accuracy (BA):** It is the mathematical mean of the real positive rates and real negative rates, offering a holistic view of model performance.

$$BA = \frac{TPR+TNR}{2} \tag{19}$$

- **F1 score (F1):** It is the harmonic mean of the true positive rate and the positive predictive value, balancing precision and recall.

$$F1 = \frac{2 \times TP}{2 \times TP + FP + FN} \tag{20}$$

- **Matthews correlation coefficient (MCC):** It gauges the effectiveness of a binary classification model, considering true and false positives and negatives.

$$MCC = \frac{TP \times TN - FP \times FN}{\sqrt{(TP+FP)(TP+FN)(TN+FP)(TN+FN)}} \tag{21}$$

- **Fowlkes–Mallows Index (FMI):** It quantifies the resemblance between observed and predicted classifications.

$$FMI = \sqrt{TPR \times PPV} \tag{22}$$

- **Informedness (BM):** It is the disparity between the true positive rate and the false positive rate, providing insight into classification model performance.

$$BM = TPR + TNR - 1 \tag{23}$$

- **Markedness (MK):** It signifies the difference between the positive predictive value and the negative predictive value, reflecting the model's predictive accuracy.

$$MK = PPV + NPV - 1 \tag{24}$$

- **Diagnostic odds ratio (DOR):** It is the ratio of the positive likelihood ratio to the negative likelihood ratio, capturing the diagnostic power of a test.

$$DOR = \frac{LdRo+}{LdRo-} \tag{25}$$

## RESULT AND DISCUSSION

Python, known for its versatility and extensive libraries, was chosen as the programming language for implementing and experimenting with the Swin Transformer model. Libraries such as Matplotlib, NumPy, TensorFlow, and TensorFlow Addons (tfa) were imported to support the development and training of the classification model. Matplotlib enabled

**Table 2 Performance metrics for different classes.**

| Metrics | Glioma | Meningioma | No tumor | Pituitary |
|---|---|---|---|---|
| TP | 131 | 125 | 53 | 108 |
| FP | 2 | 7 | 5 | 0 |
| TN | 291 | 295 | 371 | 323 |
| FN | 7 | 5 | 2 | 0 |
| TPR | 0.95 | 0.96 | 0.96 | 1.00 |
| TNR | 0.99 | 0.98 | 0.99 | 1.00 |
| PPV | 0.98 | 0.95 | 0.91 | 1.00 |
| NPV | 0.98 | 0.98 | 0.99 | 1.00 |
| FNR | 0.05 | 0.04 | 0.04 | 0.00 |
| FPR | 0.01 | 0.02 | 0.01 | 0.00 |
| FDR | 0.02 | 0.05 | 0.09 | 0.00 |
| FOR | 0.02 | 0.02 | 0.01 | 0.00 |
| LdRo+ | 139.07 | 41.48 | 72.47 | undefined |
| LdRo- | 0.05 | 0.04 | 0.04 | 0.00 |
| PT | 0.08 | 0.13 | 0.11 | 0.00 |
| TS | 0.94 | 0.91 | 0.88 | 1.00 |
| Pe | 0.32 | 0.30 | 0.13 | 0.25 |
| AC | 0.98 | 0.97 | 0.98 | 1.00 |
| BA | 0.97 | 0.97 | 0.98 | 1.00 |
| F1 | 0.97 | 0.95 | 0.94 | 1.00 |
| MCC | 0.95 | 0.93 | 0.93 | 1.00 |
| FM | 0.97 | 0.95 | 0.94 | 1.00 |
| BM | 0.94 | 0.94 | 0.95 | 1.00 |
| MK | 0.96 | 0.93 | 0.91 | 1.00 |
| DOR | 2,722.93 | 1,053.57 | 1,966.30 | undefined |

powerful visualization capabilities for analyzing the model's performance, while NumPy facilitated numerical computations and array manipulations. TensorFlow, the underlying machine learning framework, was utilized through its high-level API, Keras, making it easier to define and train the model. TensorFlow Addons extended the capabilities of TensorFlow by providing additional functionalities such as custom loss functions, activation functions, and optimizers.

In Table 2, the performance metrics offer a comprehensive overview of the model's classification accuracy and effectiveness across different classes. The model excelled in correctly classifying instances as positive, with TP values of 131, 125, 53, and 108 in classes 1, 2, 3, and 4, respectively. The model also misclassified only two, seven, and five instances as positive in these classes except class 4, respectively.

The model also performed well in correctly identifying negative instances, with TN values of 291, 295, 371, and 323 in Classes 1, 2, 3, and 4, respectively. The model misclassified only seven, five, two, and zero instances as negative in these classes, respectively, indicating a high sensitivity to actual positive instances.

The TPR and TNR metrics provide further insights into the model's performance. The TPR values were consistently above 95% across all classes, indicating the model's overall capability to accurately detect positive instances. Similarly, the TNR values remained strong with values above 98% in all classes, indicating the model's proficiency in differentiating between true negative instances and positive instances.

The PPV and NPV metrics evaluate the accuracy of positive and negative predictions, respectively. The model achieved high precision in class 1 and class 4, with values exceeding 98%, suggesting that the majority of its positive predictions were accurate. In class 2 and class 3, the model's precision was still noteworthy at around 95% and 91% respectively. The NePeVe, reflecting the accuracy of negative predictions, was consistent across all classes, with values ranging from 98% to 1.00%. This implies that the model's negative predictions were generally reliable.

FNR indicates the proportion of actual positives that were incorrectly classified as negative. It ranges from 0.00 to 0.05, with lower values being better. FPR indicates the proportion of actual negatives that were incorrectly classified as positive. It ranges from 0.00 to 0.02, with lower values being better.

The FDR and FOR metrics provide insights into prediction errors. The FDR, which represents the proportion of predicted positives that are not truly positive, was relatively low in class 4 but higher in the other classes, indicating potential areas of improvement. The FOR, on the other hand, was particularly high in class 1 and 2, suggesting that a significant portion of actual negative instances was incorrectly predicted as positive. This could be an area of concern for applications where minimizing false positives is crucial.

The LdRo+ and LdRo- metrics delve into the relationships between test outcomes and the presence or absence of the condition. The LR+ was high, particularly in class 1, indicating that a positive test outcome was strongly associated with the presence of the condition. Conversely, the LR- was relatively low, suggesting that a negative test outcome was moderately associated with the absence of the condition. These ratios provide insights into the diagnostic value of the model's predictions.

The PT reflects the point at which positive and negative predictive values are balanced. The PT values were relatively low across all classes, indicating that the model's predictions leaned more toward positive classifications. This suggests that the model might be conservative in making positive predictions, potentially leading to missed opportunities for true positives.

The TS and F1 metrics combine precision and recall to assess overall classification performance. The TS values were consistently above 88% in all classes, indicating a strong combined accuracy of positive predictions. The F1 score, which balances precision and recall, showed values above 0.94 across classes, suggesting a good trade-off between precision and recall in the model's predictions.

Pe metric is the ratio of true positives to the total number of positive classifications. It ranges from 0.13 to 0.32, with higher values indicating that the test is more precise in identifying true positives. AC is the proportion of all cases that were correctly classified. It is 0.97 for all tumor types, indicating that the test has a high overall accuracy. BA is the

average of the true positive rate and the true negative rate. It is 0.97 for all tumor types, indicating that the test performs well on both positive and negative cases.

The MCC assesses overall classification performance, considering both true and false positives and negatives. The MCC values were commendable across all classes, with values ranging from 0.93 to 1.00. These values indicate the model's effectiveness in capturing the overall accuracy of its predictions.

The FM, BM, and MK metrics provide insights into the relationship between observed and predicted classifications, as well as the predictive accuracy of the model. The FM index values were consistently high across classes, suggesting a strong resemblance between observed and predicted classifications. The informedness values ranged from 0.94 to 1.00, indicating the model's ability to distinguish between positive and negative instances. The markedness values were also high, reflecting the model's positive predictive accuracy.

Finally, the DOR captures the diagnostic power of the model's test. The DOR values were notably high across classes, indicating that the model is performing well on multiple classes.

## Visual depiction of model's performance

The model's accuracy underwent comprehensive assessment and visualization *via* a training and validation accuracy graph depicted in Fig. 3. Within this graph, the blue line denotes the accuracy progression of the training set, while the orange line signifies the model's accuracy concerning the validation set. Initially, the training accuracy commenced slightly above 0.90 on the vertical axis and showed a gradual ascent across epochs. It eventually reached a stable phase marginally below 0.98, suggesting a consistent training pattern where the model's accuracy improved and then plateaued. Conversely, the orange line, representing validation accuracy, commenced at 0.96 on the $y$-axis. It demonstrated a more variable trajectory characterized by fluctuations, signifying the model's varying efficiency on the validation set during training. These fluctuations showcased moments of higher accuracy interspersed with periods of comparatively lower accuracy.

Figure 4 illustrates the model's loss dynamics. The blue line tracks the training loss, while the orange line showcases the validation loss. Initially, the training loss starts at its highest value on the $y$-axis and progressively diminishes across epochs, steadily converging towards a specific value. Around a point marginally below 0.4, the training loss levels off, forming a horizontal plateau parallel to the $x$-axis, denoting convergence. Contrastingly, the orange line, denoting validation loss, initiates at 0.47. It portrays a more fluctuating pattern, characterized by peaks, signifying fluctuations in the model's performance on the validation set throughout the training process.

The confusion matrix illustrates the model's performance in classifying different tumor types, highlighting strengths and areas needing improvement. It showcases the model's ability to differentiate between various classes while indicating potential focus areas for enhancing classification accuracy, especially in classes with higher false predictions. In Fig. 5, the model shows robust performance in correctly identifying glioma and pituitary tumors, with very few false predictions. Meningioma and ''no tumor'' classes have slightly higher false predictions, indicating some misclassification. Overall, there are minimal false

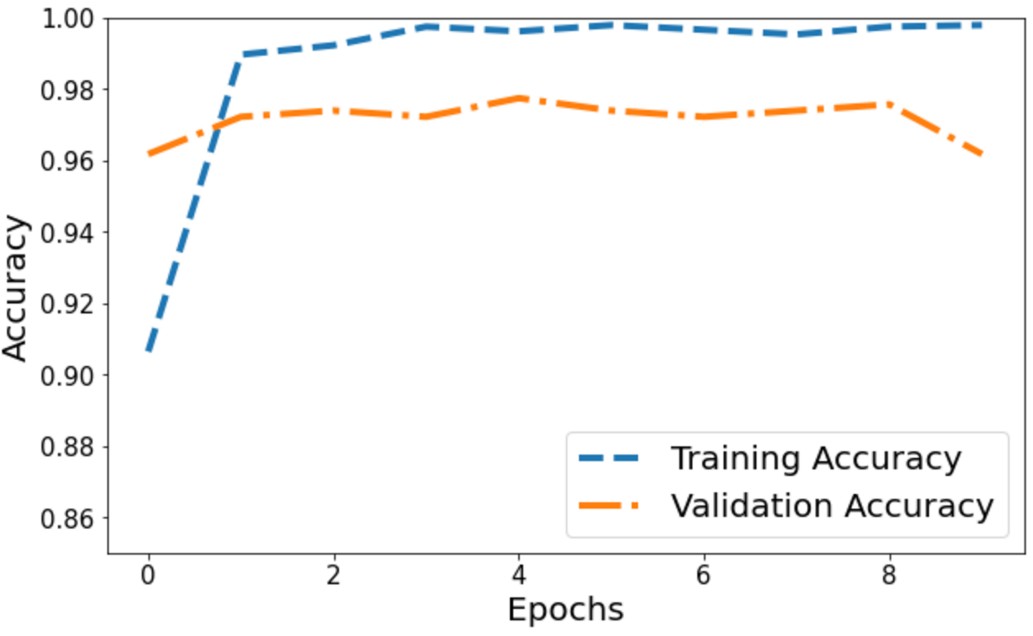

**Figure 3** Training and validation accuracy of the proposed Swin Transformer.

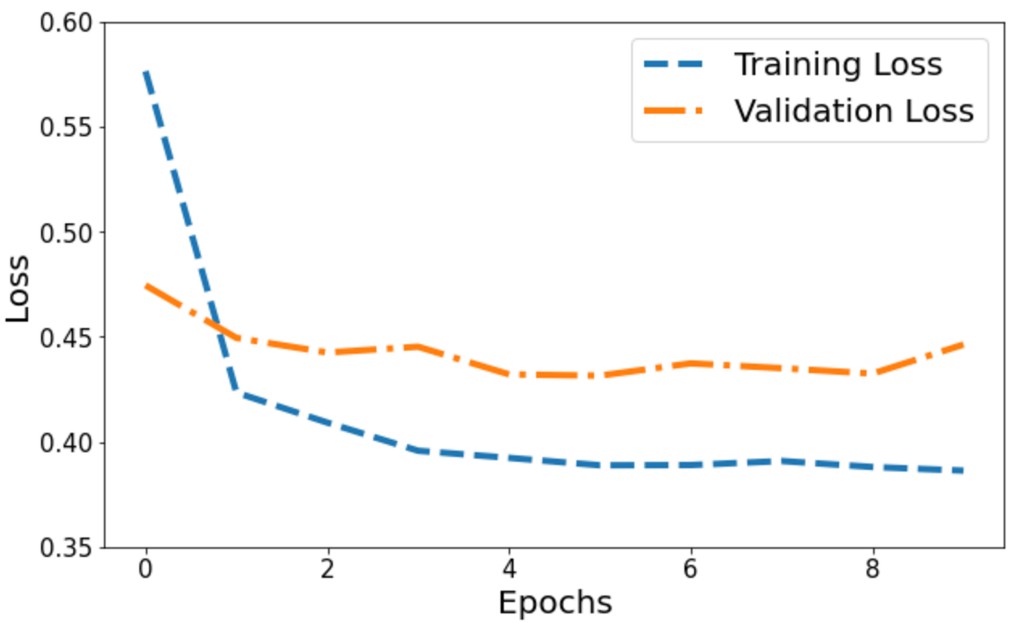

**Figure 4** Training and validation loss of the proposed Swin Transformer.

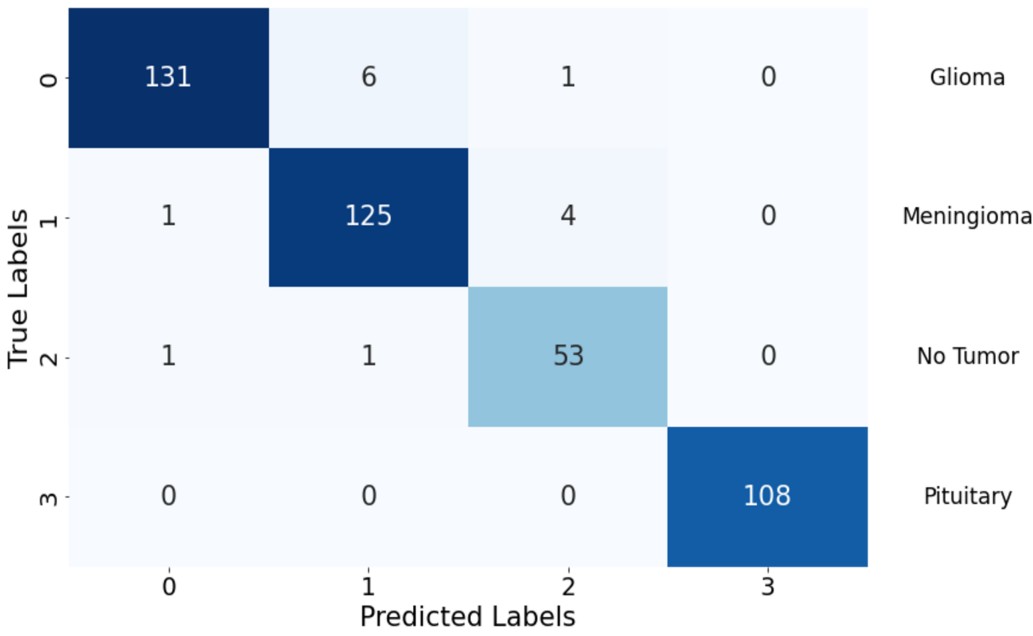

**Figure 5** Confusion matrix of the proposed Swin Transformer.

negatives (FN) across all classes, suggesting the model's sensitivity in detecting actual positive instances.

Correctly classified images: Fig. 6 showcasing sample images correctly classified by the model adds credibility to its accuracy. Highlighting these instances demonstrates the model's capability to accurately identify distinct tumor types. Incorrectly classified images: Fig. 7 presents images that the model misclassified and provides valuable insights into its limitations. Understanding where the model faltered can guide improvements or adjustments in future iterations. By incorporating visual examples, we are not just presenting numbers but providing real-world instances that enhance the audience's understanding of the model's capabilities and potential areas for enhancement.

In Table 3, we evaluate the effectiveness of our proposed Swim Transformer algorithm in the context of brain tumor classification, alongside several established methods. The metrics under consideration primarily revolve around accuracy, a crucial factor in medical image analysis. Swin Transformer exhibits remarkable performance, achieving an accuracy of 97%, outperforming the other methods presented in the table. Notably, the Swin Transformer surpasses the accuracy of CNN, Deep-CNN, Fused-based ensemble methods, deep neural networks, CNN and NADE, ViT model, ensemble CNN, BMO-ViT, and Ensemble XG-Ada-RF model by a significant margin. The distinguishing factor lies in the innovative training techniques and model adaptations incorporated in our implementation, enabling superior performance compared to prior studies. Specifically, our model's unique approach to feature extraction and inter-window connectivity fosters heightened accuracy in discerning intricate patterns within brain tumor images. This outcome underscores the Swin Transformer's potential as a formidable tool, promising substantial advancements

True: glioma_tumor
Pred: glioma_tumor

True: no_tumor
Pred: no_tumor

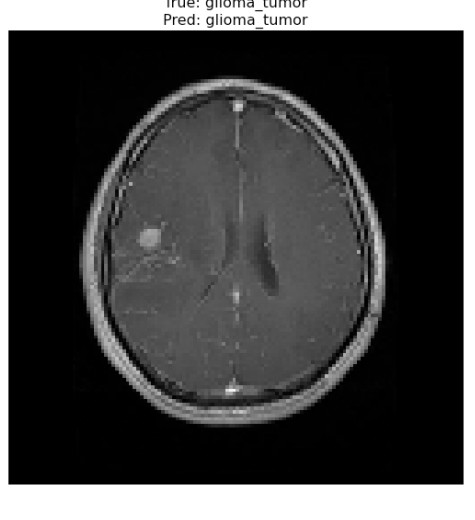

True: meningioma_tumor
Pred: meningioma_tumor

True: glioma_tumor
Pred: glioma_tumor

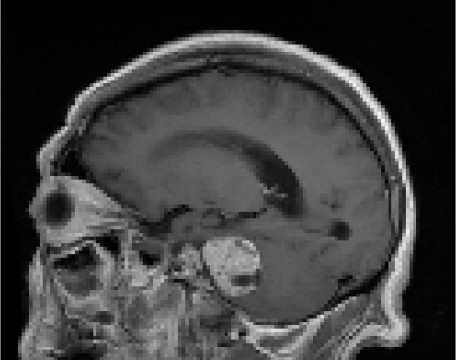
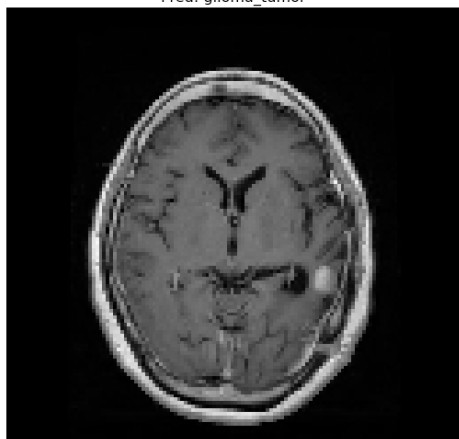

**Figure 6** **Correct classification of the proposed Swin Transformer.**

True: no_tumor
Pred: meningioma_tumor

True: glioma_tumor
Pred: meningioma_tumor

True: meningioma_tumor
Pred: glioma_tumor

True: no_tumor
Pred: meningioma_tumor

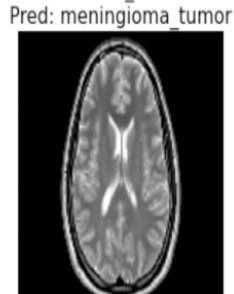
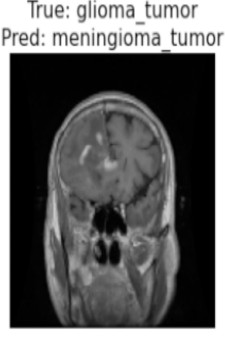
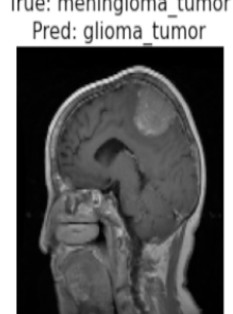
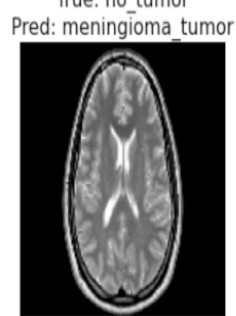

**Figure 7** **Incorrect classification of the proposed Swin Transformer.**

**Table 3  Comparison with existing work.**

| Reference | Proposed algorithm | Accuracy |
|---|---|---|
| *Amin et al. (2018)* | Fusedbased ensemble method | 86% |
| *Mallick et al. (2019)* | Deep Neural Network | 89% |
| *Badža & Barjaktarović (2020)* | CNN | 95.40% |
| *Hashemzehi et al. (2020)* | CNN and NADE | 95% |
| *Bodapati et al. (2021)* | Deep-CNN | 95.23% |
| *Nallamolu et al. (2022)* | ViT Model | 93.48% |
| *Isunuri & Kakarla (2023)* | Ensemble CNN | 96.95% |
| *Khan et al. (2023)* | Ensemble XG-Ada-RF model | 95.9% |
| *Şahin, Özdemir & Temurtaş (2024)* | BMO-ViT | 96.6% |
| Proposed | Swin Transformer | 97% |

in brain tumor detection accuracy critical for precise medical diagnostics and improved patient care. By showcasing superior accuracy and reliability compared to existing methods, our research demonstrates the distinctive contributions and advancements brought forth by our Swin-Transformer implementation.

# CONCLUSION

In our study, we introduce a novel approach for multi-class brain tumor detection and classification on MRI images, leveraging the Swin Transformer architecture. This architecture, incorporating patch splitting and merging modules, exhibited efficient processing and analysis of RGB input images. Utilizing self-attention mechanisms within non-overlapping windows, the model adeptly captured long-range dependencies, showcasing remarkable scalability. Across 21 evaluated matrices, precision, recall, F1-score, and accuracy emerge as pivotal indicators. The model demonstrates exceptional performance across diverse tumor classifications, exhibiting an impressive average F1-score of 0.96 through macro averaging and a weighted average of 0.97. In specific tumor categorizations, it excels consistently, achieving a precision of 0.98 for glioma, 0.95 for meningioma, 0.91 for no tumor, and a perfect precision of 1.00 for pituitary. Additionally, the model showcases robust recall metrics, scoring 0.95 for glioma, 0.96 for meningioma, 0.96 for no tumor, and a flawless 1.00 for pituitary. The exceptional performance extends to F1-scores, attaining 0.97 for glioma, 0.95 for meningioma, 0.94 for no tumor, and a perfect score of 1.00 for pituitary. This comprehensive performance, coupled with an accuracy rate of 97%, underscores the model's ability to reliably and accurately classify various tumor types while maintaining a strong balance between precision and recall across the dataset. Comparing our proposed model with existing algorithms, we observed that the Swin Transformer outperformed traditional machine deep learning, and vision transformer approaches in terms of accuracy and performance matrices. Future work may involve further optimizing the model, exploring additional datasets, and investigating the Swin Transformer's applicability in other medical image analysis tasks.

### Funding

This work was supported by the Deanship of Scientific Research, Najran University, Kingdom of Saudi Arabia under the Distinguished Research funding program grant code number (NU/DRP/MRC/12/28). The funders are involved in study design, problem formulation, conceptualization, analysis, paper writeup and decision to publish the article.

### Grant Disclosures

The following grant information was disclosed by the authors:
The Deanship of Scientific Research, Najran University, Kingdom of Saudi Arabia: NU/DRP/MRC/12/28.

### Competing Interests

The authors declare there are no competing interests.

### Author Contributions

- Abdullah A. Asiri conceived and designed the experiments, performed the computation work, prepared figures and/or tables, and approved the final draft.
- Ahmad Shaf conceived and designed the experiments, analyzed the data, prepared figures and/or tables, and approved the final draft.
- Tariq Ali conceived and designed the experiments, analyzed the data, prepared figures and/or tables, and approved the final draft.
- Muhammad Ahmad Pasha conceived and designed the experiments, performed the computation work, prepared figures and/or tables, authored or reviewed drafts of the article, and approved the final draft.
- Aiza Khan conceived and designed the experiments, prepared figures and/or tables, authored or reviewed drafts of the article, and approved the final draft.
- Muhammad Irfan conceived and designed the experiments, analyzed the data, performed the computation work, prepared figures and/or tables, and approved the final draft.
- Saeed Alqahtani performed the experiments, authored or reviewed drafts of the article, and approved the final draft.
- Ahmad Alghamdi performed the experiments, analyzed the data, performed the computation work, authored or reviewed drafts of the article, and approved the final draft.
- Ali H. Alghamdi performed the experiments, prepared figures and/or tables, authored or reviewed drafts of the article, and approved the final draft.
- Abdullah Fahad A. Alshamrani performed the experiments, authored or reviewed drafts of the article, and approved the final draft.
- Magbool Alelyani performed the experiments, performed the computation work, authored or reviewed drafts of the article, and approved the final draft.
- Sultan Alamri performed the experiments, authored or reviewed drafts of the article, and approved the final draft.

## Data Availability

The code is available at GitHub and Zenodo:

- https://github.com/iamshaf/swin-trans/blob/main/peerj-swin-transformer-submission.ipynb

- iamshaf. (2023). iamshaf/swin-trans: Swin-2023 (Swin-2023). Zenodo. Available at https://doi.org/10.5281/zenodo.10413239.

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
