# Peer review of "Advancing brain tumor detection: harnessing the Swin Transformer’s power for accurate classification and performance analysis"

_PeerJ Computer Science, doi:10.7717/peerj-cs.1867_

## Round 0.1 · original submission · Major Revisions

Reviewers find merits in the paper and have suggested necessary changes. You are required to address all the comments of the reviewers and resubmit paper after revision.

**Language Note:** The review process has identified that the English language must be improved. PeerJ can provide language editing services - please contact us at copyediting@peerj.com for pricing (be sure to provide your manuscript number and title). Alternatively, you should make your own arrangements to improve the language quality and provide details in your response letter. – PeerJ Staff

·

Basic reporting

The article is sufficient in introduction and background to demonstrate how the work fits into the broader field of knowledge and it is relevant prior literature.

Experimental design

The submission have clearly defined the research question, which must be relevant and meaningful. The knowledge gap being investigated should be identified clearly.The investigation have been conducted rigorously and to a high technical standard. The research have conducted in conformity with the prevailing ethical standard.

Validity of the findings

The data on which the conclusions are made available in an acceptable discipline-specific repository. The data is robust, statistically good.

Additional comments

It is observed that from table 3, the existing GA-SVM algorithm is only 0.93% less accurate than your proposed model of Swin-Transformer. Can you justify it?

·

Basic reporting

I found the subject of study quite fascinating. It is quite striking. Congratulations to the researchers. However, the study needs to be improved in terms of the following issues.
1- There is unnecessary information in the abstract, there is nothing necessary. We expect a scientific manuscript presenting the results of the study, not a report summarizing a study. I think we should see the following information in a sentence in the abstract.
1.1. What was done in this study?
1.2. Why was this study done?
1.3. What results were obtained in this study?
1.4. To what extent is the obtained result successful compared to the equivalent studies?
The abstract section should answer the questions above. Especially the success performance in the last item should be given.

Experimental design

2- The success achieved in the study and the reasons why this success was achieved should be explained in detail in a paragraph. Its superiority over similar studies must be demonstrated.

3- The organization and structure of the article should be revised to make the subject more understandable.

Validity of the findings

4-In the conclusion section, the summary section should not be repeated, but the results of this study and the success achieved should be summarized in just a few sentences.
5- I think that Result, Discussion and Conclusion sections should exist separately for such an article.

Additional comments

6- Current references on the subject should be added. The number of references is not small. However, in this type of study, I think there should be much more up-to-date references.
7- I think that Result, Discussion and Conclusion sections should exist separately for such an article.
8- Performance criteria that show the success of the work should be emphasized more clearly.
9- Authors are advised to add latest citations.
10- Overall, there are still some minor parts that the authors did not explain clearly. Some additional evaluations are expected to be in the manuscript as well. As a result, I am going to suggest Major revision of the paper in its present form.

·

Basic reporting

The references are not complete.
The novelty of the research is uncertain, as there are many research papers that have already utilized the Swin Transformer architecture for brain tumor detection. The authors should include references to these research studies and highlight the variations between them.

Experimental design

Only one algorithm is written.
The algorithm is not written in a standard and understandable way.
In Figure 2, the authors use SWIM instead of SWIN.

Validity of the findings

The references are not complete.
The novelty of the research is uncertain, as there are many research papers that have already utilized the Swin Transformer architecture for brain tumor detection. The authors should include references to these research studies and highlight the variations between them.

---

## Round 0.2 · Minor Revisions

There are still minor changes required before accepting your manuscript.

1) Algorithm 1 presented on page no. 6 has not been cited within the text.

2) On page 4-5, subheading numbers start with 0.1, 0.1.1., etc. It may be numbered as per journal guidelines.

3) In the result and discussion section, page 8, True Positive is usually abbreviated as TP, similarly True Negative as TN, and so on. True Positive Rate as TPR, and so on. Kindly correct these throughout the manuscript. Also, the evaluation metrics presented here need to be moved to the end of the methodology section.

4) The performance of the proposed algorithm needs to be compared with other existing state-of-the-art methods.

5) It is recommended to put your code on Github, or other public repository and share its link in the paper.

·

Basic reporting

This article complied with sufficient introduction and background to demonstrate how the work fits into the broader field of knowledge and relevant prior literature is appropriately referenced.This article is written in simple English and clear, unambiguous, technically correct text. The article also conformed to the professional standards of courtesy and expression

Experimental design

The knowledge gap being investigated and well identified, and statements made clearly, how the study contributes to filling that gap.

Validity of the findings

Novelty is there in research work with clear figures and tables.The data on which the conclusions are based were provided or made available in an acceptable discipline-specific repository....

---

## Round 0.3 · accepted · Accept

The manuscript has been revised as per the reviewers' and the editor's comments. I am happy to accept it for the publication in the current form.